# Public Management, Private Management and Collective Action in the Portoviejo River Basin: Visions and Conflicts

**Joaquin Romano** [1],* and **Byron V. Coral** [2],*

1   Department of Applied Economics, University of Valladolid, 47002 Valladolid, Spain
2   Faculty of Administrative Sciences, Universidad Laica Eloy Alfaro de Manabí (ULEAM), 130214 Manta, Ecuador
*   Correspondence: romano@eco.uva.es (J.R.); byron.coral@uleam.edu.ec (B.V.C.)

**Abstract:** Agricultural policies show an orientation in the management of natural resources, such as water, towards specialized production for world markets. This is promoting models of private use against those of common use. The objective of this research is to evaluate the transformations in the institutional framework associated with the change of vision of water and the pressures created on peasant communities that culturally maintain socio-ecological systems. Based on Ostrom's methodological proposals for the governance of common goods, a case study of the Rio Portoviejo Basin (Ecuador) was carried out. The three developed management models are analyzed: public, private and community. Evidence is provided that the community model is more equitable, efficient and sustainable. The way in which the extension of the market model, which conditions agricultural activity to profitability, is weakening the networks of peasant communities is also studied. In this context, the correlation between the loss of the traditional agrarian culture and the environmental degradation of the area is observed.

**Keywords:** water management; common goods; Ecuador rural landscape; collective action; rural sustainability

## 1. Introduction

In Ecuador, as in the rest of Latin American countries, there is an important tradition regarding the collective management of goods that are neither public nor private, but communal [1]. Of these assets of common heritage, water systems, food production systems, medicines, textile fibers, materials for house building, and environmental conditions stand out. These are managed with the knowledge passed down from generation to generation. Water management in the Ecuadorian territory of Manabí predates the Spanish colonization. The influence of the Inca culture is observed, which based its economy on the intensive agriculture of products such as corn, beans and livestock, and developed the construction of large cultivation terraces and stone ditches [2]. For them, the value of water does not respond to a market criterion in which sellers and buyers determine a price. Water in indigenous and small farming communities is a heritage and fulfills a social function as an integrating element of nature and society. It has cultural connotations, given that the aforementioned communities consider water as a living being and force of the universe, and has religious connotations because water, for them, in its origin, comes from Viracocha, creator god of the universe, who fertilizes the Pachamama (mother earth) and allows the reproduction of life. In ancestral communities, it is very common to exercise the right to water of the rivers, streams and channels that pass through their communes without the obligation to pay for it, taking into account the principle that water belongs to everyone in

general and to none in particular. Moreover, it also fulfills environmental functions. Water's behavior responds to natural laws and constitutes the most important value that humanity has [3].

Studies conducted through approximately 5000 cases in native communities, including high-mountain meadows in Japan and Switzerland, water projects in the Philippines and California, and fisheries in Canada and Turkey [1], reveal that the culture of water and land is the community heritage of peoples in all corners of the world. The common conclusion to all of them is the fact that for indigenous communities, water is life. In Quechua language, the term Sumak Kawsay is used, which encloses a respectful knowledge of nature. The concept of the Sumak Kawsay allows to recover a vision or perception of nature, "without disregarding technological advances or advances in productivity, but rather projecting them within a new contract with nature, in which society does not separate from it, nor considers it as something external, or as a threat, or as the radical Other, but as part of its own dynamics, as the foundation and condition of possibility of its future existence" [4] (p. 261).

The ties between the community as a form of government and the management of natural resources, and in particular, water, as common goods, have consolidated inseparable ties due to history, culture and human rights. According to Aguilera Klink et al., "Communal property has never been alien to the rules that the community members endowed themselves with so it constituted and continues to constitute the adequate solution for the life of many communities whenever it is possible to cooperate between users and provided that they do not have to face violence from governments and big private interests" [5] (p. 56).

In many of these communities, their organization and governance have, as a principle, the recognition of common goods or common pool resources, as known in the theory of the government of the commons and Public Choices [6]. The sustainable management of common goods represents their vital space. The well-being of the communities depends on the way in which they are related to the climate, water, land, lakes, forests, flora and fauna, and, in general, to all forms of life, however small or large they may be. Millions of Common Property Resources (CPR) users in the Andean countries are engaged in non-mechanized agriculture and constitute the fundamental basis of food supply and the guarantee of food sovereignty by conserving much of the diversity of food. Approximately 80% of farmers in Africa, and between 40% and 60% of Latin America and Asia are part of these structures of communal agriculture, agricultural heritage of humanity and future of man and life [7].

Associated with these agricultural practices are problems of access to water and the means to manage irrigation systems, largely administered, maintained and operated on a community basis. Many of them, despite being backed by customary rights, are subjected to great pressures to strip them and transfer them to private interests under the protection of state regulations that deny livelihoods, water rights and local management rules [3]. These causes justify the study of water resource management models in the basin of the Portoviejo River, which are developed from the intervention of the State through agrarian policy.

Concordant with the vision of good living typical of the Andean worldview [4], other elements related to culture, motivations and recognitions are incorporated. Felber calls these elements the nuclei of the economy for the common good, which seeks to resolve the conceptual contradictions about values between economy and society, highlights a scale of values around good human relations, appeals to a spirit of constitutional sovereignty, and emphasizes that economic success should become an indicator of social utility. When the exchange value is measured, reference is made to quantities and forms. When talking about utility, reference is made to what really works. The economy for the common good proposes to measure what counts, that is, what the human being needs mainly to feel satisfied and happy [8]. The result of the successful management of common goods, by organizations or communities that collectively manage the commons in a national economy, far exceed the traditional indicators of GDP that seek to measure development [9].

On the other hand, theories that advocate keeping the State away from its spaces of institutional realization (the self-regulatory market) proliferate. However, Ostrom establishes a link between them, showing that institutions are rarely either private or public (the market or the State). This intuitive

understanding, that the State maintains its institutional function as a regulator and supervisor entity of the private sector, is diminished with the case illustrations, which demonstrate that institutional support for the market is provided by the State through its regulations which provide the necessary guarantees for the market to function. In addition, as Ostrom states, "No market can exist for long without underlying public institutions to support it [ . . . ] public and private institutions frequently are intermeshed and depend on one another, rather than existing in isolated worlds" [1] (p. 15). This facilitates a framework that disturbs the possibility of finding different solutions to the problem of the sustainability of natural resources. This last one is understood as the ability to apply extraction processes to the components of the ecosystem in such a way that they preserve their structural and functional characteristics, such as those raised in the theory of collective action.

This rupture of the State-market dilemma for the inclusion of common goods is supported by the legal sciences, since from the point of view of Law, the practice of common goods is found without much difficulty in the traditional management of communal goods. Such custom still persists in Europe and is also present in the traditional law of the indigenous communities of Latin America [10]. In Roman law, there was a consideration of the so-called "res communes omnium", the common goods of all (air, running water, the sea and its shores), but legal-political measures were not necessary to preserve the "common things" and guarantee its enjoyment by all [11]. Nowadays, new categories of common goods emerge, linked to environmental protection, such as water or the atmosphere, but also to other areas such as outer space, the Antarctica or the oceans. As has been said, the premises of current legal systems are based on a nineteenth-century individualism, which have possessive individualism as a cornerstone without considering other factual realities as not only possible but materially existent [12]. The "anthropological ignorance" of the jurists has been spoken about to show that current law and many jurists are not capable of conceiving a property different to the one of the dogmatic 19th century civil law [13].

However, an increasingly broad current welcomes jurists who have become aware of the reality of common goods as an autonomous legal category and the need for their express protection by Law. Probably its most qualified theoretical representative is the Italian, Ugo Mattei, who has stated that "Awareness of common goods as political and constitutional instruments of direct satisfaction of the needs and fundamental rights of the community does not arise from offices. Rather, it constitutes a political product, still technically amorphous, that takes root in the deep sense of injustice that gives life to Law" [14] (pp. 11–12).

In this author's position, the factual reality of common goods requires the construction of categories and institutions that account for their existence and allow their preservation and protection, in order to exclude them from both commodification through private property and its exploitation by the State. Furthermore, common goods are always linked to a community (they serve their common good), thus, their contextualization in the community environment is essential. As a characteristic, common goods respond to their use value in the community and not to their exchange value in the market, being accessible to all. Accordingly, they must be treated as goods outside the market that play an essential role for everyone in the community.

The globalizing dynamics extend the great transformation of rural landscapes, associated with the privatization of natural resources. Such resources are traditionally managed as public or common goods. The effect of market globalization on rural communities, and particularly on small farmers, in Latin America, is an important topic in the literature [15,16]. Development policies in Portoviejo incorporate the extended vision of the country progress and the well-being of the peoples. This well-being depends on the productivity and export capacity of agricultural products as indicators of competitiveness and human talent based on the efficiency of the use of natural resources. The aim is to determine the impact on Socio-Ecological Systems (SES) and social capital [17,18], derived from changes in the management model of water resources in the basin of the Portoviejo River, located in the canton of Portoviejo in the Province of Manabí (Ecuador) (Figure 1).

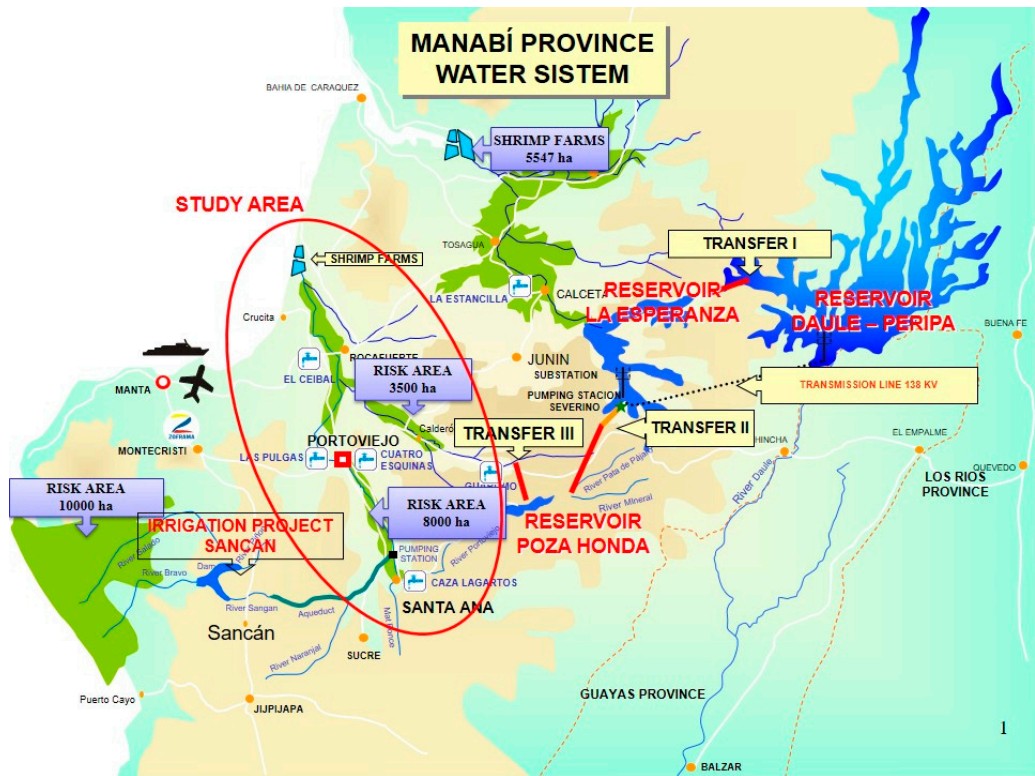

**Figure 1.** Map of the hydrological system of Manabí and study area of the Portoviejo River.

The Portoviejo canton has an extension of 967 km$^2$, which represents 5.12 percent of the surface of the province of Manabí. As of 2014, it had an estimated population of 304,227, with 80 percent of inhabitants under 50 years of age, of which 17,294.877 live in urban parishes and 66,583 in rural areas. The active population is 108,353 inhabitants, of which 62 percent is employed by the tertiary sector, while agriculture employs 16 percent [19]. Statistics for rural areas are imprecise, given the predominance of the informal economy.

The hydrological system is characterized by having various water sources which supply water to the population for human consumption and irrigation water for agricultural production, as can be seen in Table 1.

**Table 1.** Flows of the hydrological system of the Canton of Portoviejo.

| Basin Name | Environmental Flow (m$^3$/s) | Water Quality |
|---|---|---|
| Poza Honda Dam Site | 3.26 | Contaminated |
| Santa Ana Diversion Dam | 7.42 | Contaminated |
| Portoviejo River in Portoviejo | 12.35 | Contaminated |
| Portoviejo River A. J. Chico | 13.74 | Contaminated |
| Riochico River in Alajuela | 3.5 | Contaminated |
| Riochico River A.J. R. Portoviejo | 6.45 | Contaminated |
| Portoviejo River-Ceibal | 21.62 | Contaminated |
| Portoviejo River Estuary | 22.19 | Contaminated |

Source: SENAGUA 2011.

From a sample of farming families, we analyze the effects of irrigation infrastructures on the vision or perception of water as a communal resource, prevalent in rural communities. The construction

of the Poza Honda reservoir on the Portoviejo River has expanded the irrigation area and the incorporation of cultivation practices of productive agriculture (Figure 2). Its impact is considered in the conservation of social networks that structure the SES, which are representative of a culture that, over time, has generated traditional ecological knowledge (TEK) [20,21], considering determinants of sustainability. The basis of this research study are the three models of water management that have occurred in the Poza Honda irrigation system.

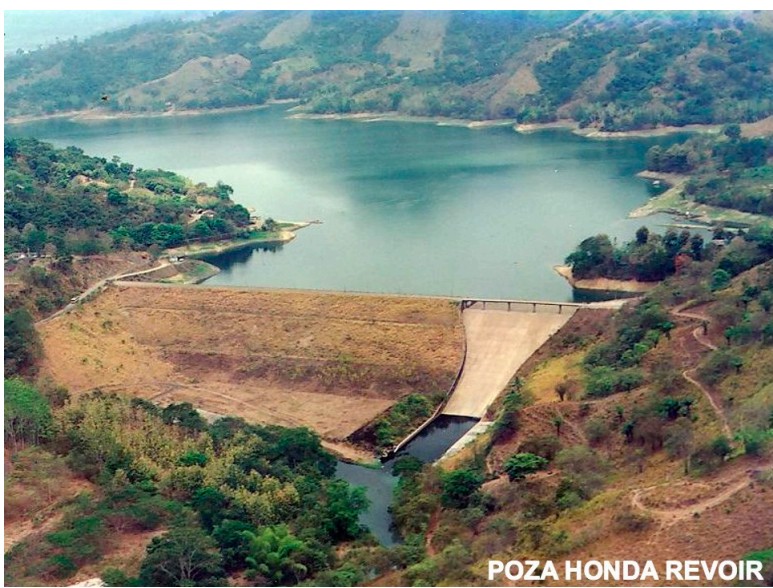

**Figure 2.** Aerial view of the Poza Honda Reservoir.

A first model was extended until the mid-20th century, in which the agricultural community in the basin of the Portoviejo River was dedicated to the production of traditional crops, closely related to the concept of food sovereignty. Starting in 1962, Manabí began a second model of public water management, as a result of a great mobilization of small farmers, workers, students, housewives, educational establishments and social organizations, who organized, in a great front of struggle, the demand of solutions to the water problem, and forced the National Congress in 1962 to create the Manabí Rehabilitation Center (MRC), whose function was the planning and execution of potable water works, irrigation and use of the province water resources. The most significant public work was built in this stage, which begins with the construction of a water reservoir called Poza Honda Dam, with a storage capacity of 100 million cubic meters of water, the initial cost of which amounted to 123.2 million dollars (at current prices of 1969). This dam came into operation in 1971. The original aim of this work was to provide water for human consumption to the inhabitants of the cantons of Santa Ana, 24 de Mayo, Portoviejo, Jipijapa, Montecristi, Manta, Rocafuerte and towns in between. Moreover, in order to take advantage of the surplus, a system of canals was built in 1978 to irrigate crops in the valley of the Portoviejo River.

A third private management model for Manabí water resources was introduced in May 2003. This model arose from the reforms brought about by the so-called Washington Consensus, the Law of Modernization of the State, Privatizations and Provision of Public Services by the private initiative, the Reform Law to the Constitutive Law of the MRC, and the MRC resolutions. These resolutions sought to transfer to the private initiative the operation, maintenance, administration and construction of the water works through the company Manageneración, which was created for this purpose.

The main objective of this research is to analyze, in terms of SES and social capital, the transformation processes that are taking place in rural communities in the context of irrigation infrastructures linked to the Poza Honda reservoir and its water management models. This research offers contributions to three basic questions that these transformations raise: Does the increasing

interference of world markets in collective decision-making processes produce a change in the vision and model of water management in rural communities as a communal good to a private good? To what extent do these causes disrupt ESS and affect social capital? What are the impacts that these transformations have on the ecological systems of the basin under study?

The case study offers results that show the division of producer typologies that globalization has introduced in the study area. Family groups that have less access to irrigation water and land ownership are in conflict with those who practice intensive and productive agriculture, with state support. As a result, the cultural degradation of the peasant communities is verified, with the loss of the TEK acquired on techniques and varieties of crops that guaranteed food sovereignty and security. The increased demand for land and water for agriculture has an ecological impact on the sensitive dry forest ecosystems of the region, and causes adaptation problems to climate change.

## 2. Materials and Methods

Methodologically, the analysis is carried out from the new institutional economics approach, designed and used by Ostrom in studies of systems of common pool resources around the world [1,18,22]. Ostrom's theoretical proposal includes the Institutional Analysis and Development Framework which includes the broadest categories of variables that can be affected in an action situation, such as:

1.  Biophysical conditions, which involve the analysis of goods according to the four categories of defined goods (common pool resources (CPR), public goods, private goods and toll goods) which are also the characteristics of the cultivated ecosystem.
2.  Attributes of the community, which include the history, culture, customs, their knowledge, techniques and social capital; the latter, understood as the set of networks of interpersonal trust between human societies, also known as the productive social system.
3.  Rules-in-Use of the system, which englobes what is allowed, what is done by obligation and what is prohibited. These rules include agreements on how the system is managed, who can act within the system, with which functions its members participate, and the type of sanctions for non-compliance with the rules, that is, the system of government.

The characteristic of complexity is introduced in the analysis of institutions and the governance of natural resources as a key element, in which the existing interrelationships between social and ecological systems are observed, leading Ostrom to propose the study of SES [22], or what Mazoyer and Roudart denominate Cultivated Ecosystem and Social Productive System [7] (p. 72), in which it is possible to observe the multiple forms of organization and operation to which natural resources are subject.

The processes of change are the result of a vision that represents the social perception or cognitive representation of the territory dynamics, and of the participation of agents and social dialogue, as has been discussed in landscape research [23]. Some analysis of the theoretical or conceptual foundations that underlie these approaches when evaluating the perceived values of the landscape observe the absence of an explicit theoretical foundation [24]. However, at the same time, the importance of understanding the perception dynamics of the human landscape and its importance for integrating and implementing it in current policy strategies is recognized [25,26]. Landscape conservation in rural areas and indigenous peoples is closely linked to the intangible heritage and the attachment or sense of place of their communities [27–29].

Ostrom methodological proposal is multidisciplinary, considering the need to address complexity through multiple methodologies from different disciplines. It is also transdisciplinary, focusing on real problems. This is reflected on the suitability of the case study, which meets the objective of solving real problems in the practice of collective action in ecological systems and landscapes that meet the characteristics of common goods [30].

The case study of the basin of the Portoviejo River is applied to the research. It is organized according to a protocol that seeks to understand complex processes. It is ideal to work with natural

resource systems that involve experiences of collective action and common goods, considering that it is the only empirical field research option when cross-data is not available [31]. The investigation began with the compilation of the available documentation of the socio-ecological system of the studied area. This documentation does not provide information on the action situation and the perception of agricultural producers. Therefore, field work has been carried out with interviews and direct observation, complemented by interviews with the authorities and other institutional actors responsible for managing the basin.

The collected information for this case study begins with an interpretation of the landscape, which helps to recognize the different elements and the biophysical conditions of the cultivated ecosystem along the basin of the Portoviejo River. This interpretation allows us to have relevant information on the infrastructures related to the water system, its state, and the way the territory is organized for agricultural production.

The study area is located, according to the Bioclimatic map of Ecuador, in a region classified by Holdridge as subtropical desert, and on the Phenology map of Ecuador as semi-deciduous [32]. The climatic characteristics of the area, with two clearly differentiated seasons, have determined the ecological systems and conditioned cultivation practices. Vegetation varies in each of these seasons, with the wet season exhibiting a predominant coverage of hydrophytes, epiphytes and climbing plants. Although practically all groups of fauna are found, birds constitute the most representative group, and only common species of medium or small size are found. Regarding mammals, these have disappeared, or their populations have been reduced in most of the territory and have taken refuge in areas of difficult access for man. In Manabí, endangered species have increased, such as agoutis, deer, oncillas, lowland pacas, partridges, guacharacas or the eared and rock pigeons.

The study area is characterized by agricultural activity, with family farms predominating. The traditional cultivation practice is one of short-cycle crops, mainly in the dry season, between June and November. The low pluviometry effect contributes to a great extent, during the dry season, to the water degradation in all the tributaries that form the Portoviejo River. In the wet season, from December to May, heavy rains cause overflows and floods that hinder crops. This shows the sensitivity of this area and requires integral management of crops to improve food security and sustainable development [33].

Regarding land use, the III National Agricultural Census shows that until 2000, most of the canton area corresponded to mountains and forests, and to transitory crops and fallow, which covered an area of 26,206 ha and 13,109 ha, respectively (Figure 3). The area of crop and pastures is estimated at 37,000 ha, which occupy almost 40 percent of its surface, from which 16 percent is pasture and 24 percent are crops. A total of 39 different varieties of crops have been identified, highlighting dry corn which occupies 39 percent of the cultivated area, coffee with 16 percent, hard corn with 7 percent, and lemon or cocoa with 5 percent. The Survey of Surface and Continuous Agricultural Production of the National Institute of Statistics and Censuses of 2018 indicates a reduction in mountains and in the crops of forests of 8 percent, and an increase in the areas cultivated with irrigation by 16 percent, mainly in dry corn. The irrigation system varies from transitory crops, in which the furrow or flood crops dominate in 86 percent of the surface, to permanent crops in which this system is 38 percent; and in 55 percent, the irrigation is by sprinkling [34].

In the next step, 27 in-depth interviews are carried out with producer families. The selection of the families was made following the non-probability sampling technique; in this case, criteria of access to land, water and the predisposition of families to provide the requested information were used. Each interview lasted between three and four hours, accompanied by field observations lasting an additional one to two hours. The interview guide used corresponded to a properly structured questionnaire to collect the information provided by producers with irrigation and without irrigation. This instrument is divided into eight parts.

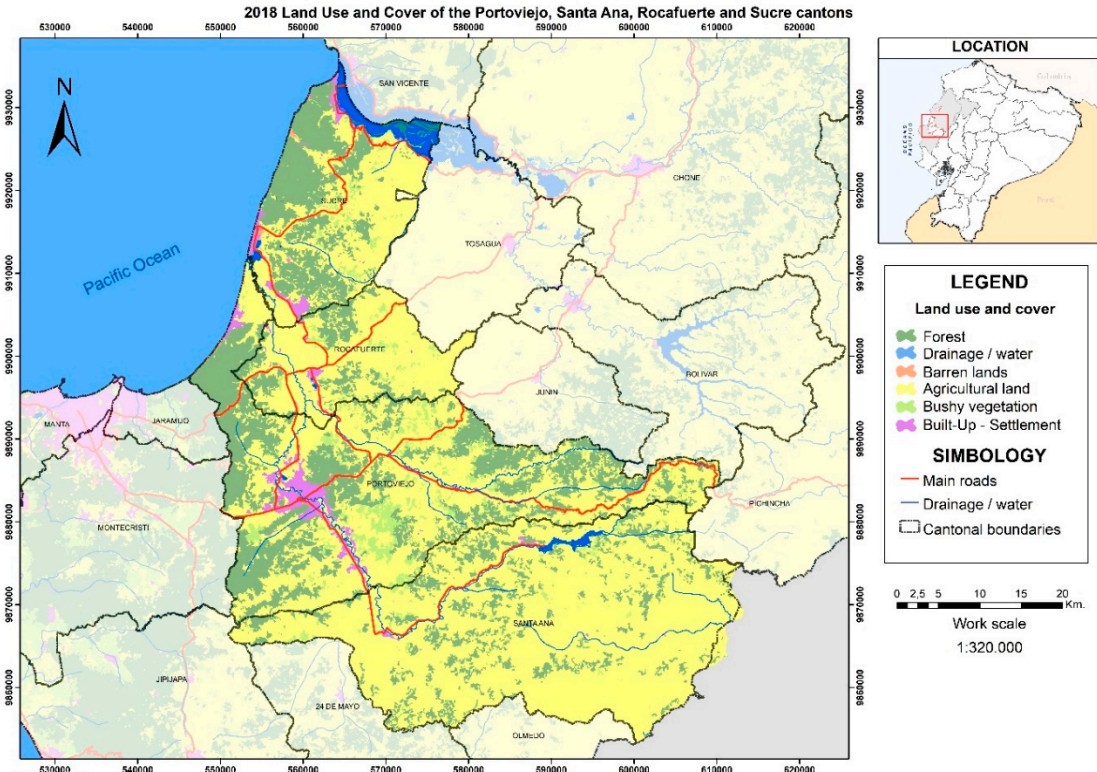

**Figure 3.** Map of land use and cover of the Portoviejo, Santa-Ana, Rocafuerte and Sucre cantons. (Source: Ministerio del Ambiente y Agua, Ecuador).

The first part collected location information on the property. The second part dealt with the composition of farming families and the time its members spend on agricultural work on their farm, in the production of other farms as wage labor and in other activities not related to agriculture. The third part reported on the income that rural families earn from work related to migration. The fourth part dealt with information on other types of income that families earn. The fifth part provided information on the characteristics of agricultural holding. The sixth part of the interview collected information on the technical and economic characteristics of the cropping systems, in some cases with information on more than two crops. The seventh part gathered information on the technical and economic characteristics of animal husbandry. Lastly, the eighth part sought to identify the resources that producers have for agricultural work. It is important to highlight that the interview guide allowed the recording of valuable information regarding the questions asked, and also made it possible to collect comments and testimonies on the relationship of the families with the environment, the market, social capital, and state entities.

Although the main source of information comes from the structured interview carried out, due to the heterogeneity of the peasant families interviewed, we have carried out semi-structured interviews as one of the instruments of the ethnographic method, given that they offer an acceptable degree of flexibility, while maintaining sufficient uniformity to achieve interpretations consistent with the purposes of the study. The information from the interviews has been completed with that obtained from observing their irrigation practices. The aim was to describe the rationalities of small farmers and the collective rule systems behind their daily practices. It was possible to understand the management and control actions related to water, soil and vegetation resources in the activity of producers.

Secondary information from official sources was also used for the analysis, so as to observe the institutional trajectories for the management of water resources in Manabí, mainly the one related to the legal reforms that modified the way of seeing and undertaking water management in the basin. In order to assess this information, nine open-ended interviews were carried out, distributed

as follows: two to researchers in the field of agrarian economy, who were familiar with the study of agrarian systems in Manabí, four to community leaders linked to the problem of irrigation in Manabí, and three to officials of the Manabí Water Resources Regulatory Corporation, a public institution in charge of managing Manabí's water system. The documentation reviewed is extensive. On the one hand, it includes Williamson recommendations in the framework of the so-called Washington Consensus that were implemented in Latin America from 1989 [35], the Law of Modernization of the State, Privatizations and Provision of Public Services by the private initiative of 1993, the Reform Law to the Constitutive Law of the MRC of 1994, the reports of minutes, agreements and contracts made by the MRC for the construction, operation, maintenance and exploitation of hydroelectric power generation plants, and the operation, administration and maintenance contracts for the pumping dams, water transfers and related works, and, in general, for the privatization of the Manabí water since 2003. In addition, the agreement was analyzed to transfer the infrastructure of the Rocafuerte Irrigation System to the community of families organized in the General Board of Users of the year 2000.

## 3. Results

### 3.1. Characterization of Agricultural Producers in the Case Study

From the case study conducted in the basin of the Portoviejo River, it is deduced that the producers interviewed develop different types of family farming with notable differences between them, both in its material and immaterial aspects. Conventionally, they have been differentiated by income levels, mainly depending on the cultivated area and the access capacity to irrigation water. However, their most relevant characteristics are linked to the cultural vision they have of their activity. In other words, the meaning they give to working with communal goods such as water and land to obtain food sovereignty, around which the government systems and social networks of peasant communities are generated and structured.

According to the information resulting from 27 interviews with the selected producers, three types of producer families have been identified. Each family type has undergone nine in-depth interviews. The first group is formed by those families whose production is more oriented to self-consumption. The income they generate is not enough to guarantee family reproduction, so they must resort to wage labor, which Soto calls Family Subsistence Agriculture. This group includes producers who do not have access to irrigation and those who have irrigation but whose surface is very small, less than two hectares. These producers must offer their wage labor force, and the rest of the family members must carry out other activities to obtain the income for their subsistence [36].

A second group, which Soto calls Family Consolidated Agriculture [36], is made up of the interviewed producers who, on average, have between four and six hectares, have access to irrigation, and the family depends, to a greater extent, on the production for their own consumption with the surplus going for sale in short chains. Although family reproduction is possible with these type of producers, they are not capable of generating the surpluses that allow the reproduction and development of the productive unit. The progress of this group depends, to a large extent, on the support they may find in public policies regarding financing, market insertion, and negotiating capacity and access to technologies to make them more efficient, without distracting them from their role as producers for food security in the region.

In third place are those families who have properties with areas greater than ten hectares and have access to the irrigation system, are able to ensure food supplies for their families, and the income they obtain from the sale of their products allows them to capitalize on the farm. Moreover, this group has a high degree of market entry, has and employs more technology, and uses agrochemicals such as fertilizers, herbicides and fungicides, which, among other effects, cause heavy contamination in the basins, especially in the dry season when the flows decrease significantly.

According to interviews with the producers, those who have had access to irrigation have achieved increases of over 200% compared to those who do not have access to the irrigation system.

The availability of irrigation for dry corn, which is the main crop in the area, allows to produce more than one cycle. However, this increase has also meant dedication to crops that are more intensive in the use of water and agrochemicals. The agrarian leaders interviewed, belonging to the first and second type of producers, do not agree to maintain agricultural policy. Particularly, subsidies are responsible for promoting a type of production that is increasingly remote from production for food security that has been characterized in the past; that is, conserving genetic material and cultivating the diversity of foods that the community wants culturally.

Of the total number of agricultural producers in the study area, estimated at about 16,000, 78 percent correspond to those of the first and second groups, but together, their production is estimated to represent 50 percent of the marketed value. The observed trend is towards an annual reduction close to 5% of those of the first type, both due to abandonment of the activity and due to their consideration as non-professional producers for self-consumption. In contrast, there is a growth of the third type of family, among other factors, due to the requirements of the public aid to agriculture

### 3.2. Water Management Models and Conflicts

Water has always played a fundamental role in the institutional framework of rural communities. The dynamics of the management models of this resource allows to identify the characteristics and evolution of their culture. In the case study, three different models have been identified. A first model corresponds to a communal water management system. It has its origin in the model of indigenous communities that practiced subsistence agriculture, with strong spiritual values towards nature [37]. They share seeds, labor, and, in many cases, production technology, and trade was very small. They practiced family farming in harmony with biodiversity and the protection of genetic resources. In general, the concept of common goods and the character of solidarity allows them easy access to productive resources and, therefore, provides sufficient and adequate food for their culture.

Currently, the abovementioned community management model is presented in a context of producers who organize to manage the resource and the irrigation infrastructure, maintaining the objective of guaranteeing access and conservation of the hydrological system, but in which production is mainly oriented to local and global markets, and to a lesser extent to self-consumption, according to the typologies described. This model for the common good management of the irrigation system began with the transfer of the irrigation system to the users as a state policy, according to the Agrarian Development Law of 1994.

The abovementioned model was part of the modernization strategy of the State along the lines of the Washington Consensus, rather than in recognition of communal forms of government with experience in the sustainable management of natural resources. The transfer was signed on 12th December 2000, through an agreement that gives detailed account of the infrastructure being transferred, although it says nothing about the state of such infrastructure. The users of the system declared in the interviews that this transfer was not carried out under the best conditions in favor of the users, organized for this purpose at the General Meeting of Users of the Rocafuerte System, on 24 March 2000. The transfer was done in a hurry, with no time to establish a communication strategy to explain and involve users in the commitment to manage their irrigation system, and without having the resources to finance the work necessary to maintain the system. Despite this, the producers assumed the challenge of organization that allowed them to maintain the system and carry out canal reconstruction or repair processes.

Currently, informants think that the most difficult part is over and that things have improved since the transfer. Users have established institutional mechanisms for the conservation and operation of irrigation infrastructures, which are financed by themselves. Moreover, although some still do not want to participate in these mechanisms, most of them use and finance them. In recent years they have built dams with pieces of wood that come from their farms; they share food, money and work. It is part of everyone contribution, which constitutes collective action work that they propose to extend to other fields of agricultural work.

The General Board of Users is in charge of the general administration, while a system of boards is responsible for organizing users. The board has a set of rules that regulate the rights and obligations of the users. They know that many of their problems have been solved, but they also know that there are still great challenges to face, related to the payment of water and to redistributive ways that could contribute to improving the general conditions of the system. This management model is characterized by a comprehensive approach in harmony with nature, as stated by the producers interviewed. Its planning is participatory considering environmental, economic and social criteria. The administration of the system is of shared responsibility, in which they have rights and obligations. For the members of the transferred system, water is priceless. However, they assign it a fundamental value for life. Thus, it has cultural, social and environmental connotations.

According to testimony from MRC officials, in an open interview, public management was characterized by the construction of large works of water infrastructure, in which water priorities had a strong component of political influence, and by technical and centralized planning, which is due to an administration with slow responses due to paperwork and interposition of functions. Social policy and some environmental concern had earned it an important appreciation on the part of the communities, which also felt that this institution was theirs because of the way it had been achieved. It is insisted that in this model, the priority is to satisfy the needs of water supply for human consumption and water for irrigation.

The second model identified is that of public management. A first stage can also be distinguished between the years 1960 and 2000, in which the State carries out an investment policy in irrigation infrastructures in the main basins of the country, and assumes the exclusive management of them. The conflicts that this model created in the primary sector, both with peasant communities and with agricultural companies, led to the application of community management and private management models. The communal model has the supervision of public authorities, which act on the basis of policies from different areas and administrative levels. Agricultural policies, development, water, protection of natural areas, etc., of the central government of Ecuador, and of the decentralized government of the provinces and cantons, establish a complex framework of public water management.

Finally, the third management model is the one of private management, which began to be implemented in 2003. In this model, the priorities in relation to the use of water consist in maintaining a constant water flow for the profitable production of hydroelectricity. Planning is focused on efficiency criteria to satisfy the demand for electrical energy. Additionally, water infrastructure and the water contained in these works are considered financial assets, incompatible with criteria of integral management of the basin that involves environmental, social and economic issues in a balanced way.

The three models of water resource management, as seen in Figure 4, have emerged at different times as a result of the predominance of the strength of social or economic groups, which, in turn, have succeeded in making their interests prevail. In this context, the most significant conflicts are observed in the priorities that each group sets of the functions that water must fulfill. According to the terms of the contract, the MRC must guarantee a continuous flow of water to Manageneración for the production of energy. This contrasts with the interests of the irrigators since it is required to regulate the flows from the dam according to the greater or lesser rainfall precipitation. The emergence of Manageneración and the absence of public water management also meant the dismantling of all social policy and technical assistance provided to the agrarian community. Other conflicts have been registered, according to information from the open interview, as a result of using the land, with an agrarian purpose, to dedicate it to shrimp production, and this has caused, according to the MCA technicians, soil deterioration and water contamination that returns to the channel without due treatment.

In these three models, the considerations of the ecosystems linked to the Portoviejo River Basin have been very different. The general appreciation is that while the community management model corresponds to a culture better adapted to the environment and has been able to better protect it, the other two models have not sufficiently taken into account the impacts on the natural environment,

and have treated water as an economic resource for private use. However, the question is not so simple. Even in the current context of communal management, the dynamics of land use shows the recession of the forest edges due to the expansion of crops and natural pastures. This has altered the forest areas of seasonally dry forests with varying degrees of intervention or pressure. The transformation of native vegetation into crops, grasslands and populated areas by burning bush vegetation and discharging pesticides into ecosystems, is the main cause of loss or threat to wildlife [38].

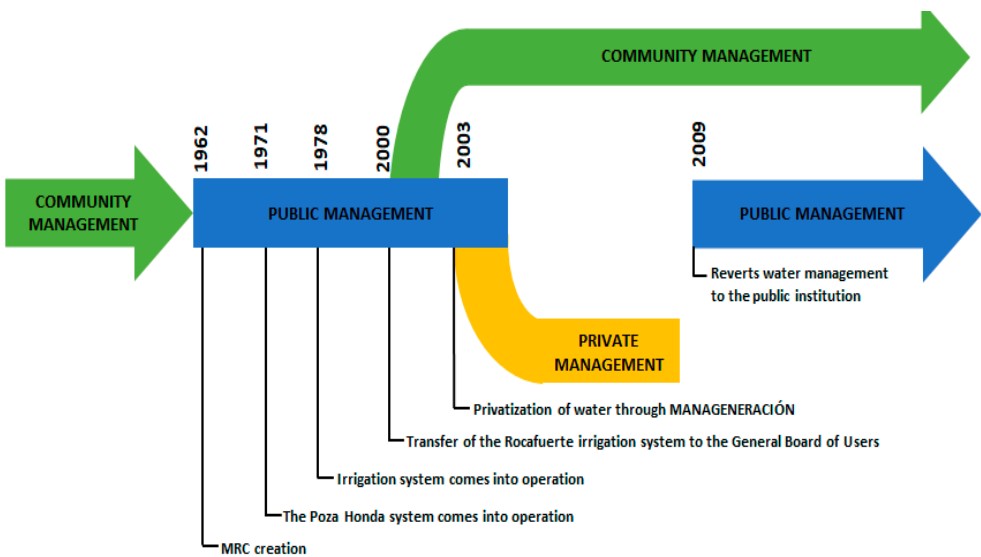

**Figure 4.** Water resource management models in the basin of the Portoviejo River.

*3.3. Contributions of an Institutional Approach in the Case Study*

The institutional frameworks differentiated in the sustainability of the SES are derived from the predominant conception of water as a private or community good. The study of the effects of irrigation infrastructures on each of the three types of producers has allowed identifying a change in perception towards profitability values introduced by the markets, to the detriment of the cultural values of small farming communities. Practically all interviewees believe that increasing the availability of water for irrigation and soil for cultivation would increase their productive capacity and proportionally, the profitability of their family farms. Paradoxically, the peasants of the first type, despite having less land, are those who show a greater concern for preserving the quality of the soil and water. The reason they offer is their dependence on family and community work. They are not alien to new agricultural technologies, but they lack the ability to access financial capital. Half of these producers say they have not been favored by the irrigation infrastructures built. On the contrary, their difficulty in exchanging or selling their productions at fair prices in short chains has increased, since the relative prices of food have decreased.

As a consequence, these families have further increased their social vulnerability, losing many of the mutual support networks they have woven, and being excluded from those created in the market. For market institutions, they are considered not very competitive, and they also suffer from a lack of public aid, which, for the most part, is conditioned by the introduction of investments in agrarian modernization. In the interviews, they do not declare a desire for their children to continue their activity, which is indicative of a loss of sense of place that is leading the youngest to abandon farming and migrate to urban areas. Nonetheless, it can be said that this group brings together the most representative TEK of traditional culture, with farming practices and management of common goods strongly linked to ecological processes and with unappreciated ecosystem services.

The farming families, in their role as guardians of varieties of seeds adapted to extreme weather, soil and water conditions is not recognized. The observed loss in crop diversity put a risk in the

conservation of a genetic material that has unique characteristics. In traditional culture, seeds have always been considered communal goods, around which fundamental social networks of TEK exchange have been woven. The replacement of these seeds by industrial varieties that are managed as private goods by large business groups that operate worldwide, is also representative of the change in the institutional framework [39,40].

## 4. Discussion

Intensive agriculture, which originates from the institutional framework in global markets, starts from the assumption, quite generalized in Western culture, that water is considered a lifeless resource. It is neither plant nor animal. It is defined according to its physical properties as a liquid mineral that has great potential in the area of production. Its importance lies in the extraction capacity of environmental products that it produces or those that are derived from it. Under this approach, international organizations, such as the World Bank, require the countries to which they lend money to treat water as a private good; thus, encouraging entrepreneurs to develop creativity to package water as a product and to assign a price to it based on the market rules of supply and demand [41].

On the other shore, small farming communities consider water as a common good. It performs environmental, social, economic, cultural and landscape functions [42,43]. There are also social movements that promote a new culture of water, which gives way to efficiency and imagination, to subsidiarity and participation in management, to the real economic, social and environmental accounts of water, and to the humanistic conception of the resource. We are all river users. The new culture of water must end with the misrepresentation of the current concepts of "demand" and "resource", with which it has been tried to establish an unreal panorama of unsustainable imbalances to justify the establishment of a great state of works that restores a hydraulic balance that nature never had before [44,45].

The abovementioned conflicting visions about the character and use of water and, in general, of common goods, are transferred to the realm of events, where occurrences can shed light on the best way to manage these generally fragile ecosystems. For advocates of centralized regulation, the State should control most of the common goods to prevent their destruction. However, a large part of the natural resources that have been intervened by the states around the world, do not have major achievements to display. The case of the Portoviejo River also confirms Ostrom thesis, that often in places where the State has succeeded in imposing its rules, the effects on the ability to agree and develop joint actions for cooperation have been null or very limited [22,46].

Investments in irrigation infrastructures in the basin of the Portoviejo have introduced significant changes in agriculture and socio-ecological systems since the 1970s. Firstly, new access conditions to water for irrigation are observed, with more exclusive and rival mechanisms that put at risk the communal good condition of this resource. Although there is an ancestral culture of communal water management in the basin of the Portoviejo River, the new conditions of provision of the resource are leading to an agriculture oriented to obtaining profitability in the markets rather than to obtaining food sovereignty.

The interviews conducted show that the irrigation infrastructures have determined a differentiated typology of producers, as well as a conflict between them. Facing an emerging family type that applies simple criteria of economic profitability, based on intensive farming and large consumption of water for irrigation, there is an opposing family type that maintains traditional cultivation practices and communal networks linked to water. A second opposing type of family incorporates improvements in farming systems and adapts community networks to local markets as a meeting place. Culturally, these groups that farm for self-sufficiency and exchange in local markets have shown to be efficient in providing food for local communities while preserving ecosystems. However, the measure of efficiency has been reduced to the value of productivity in the markets. National accounting associates the short- and medium-term growth of primary sector productivity with farmers who direct all their production to commercialization in the markets. This has lead to the widespread conclusion that in Latin America,

capital has been the most important productive factor in explaining increases in output, and not work or land factors [47].

Despite the fact that studies indicate that the highest yields and profitability of these cultivation models are only maintained in the short term, State intervention is favoring productive agriculture in the conflict, and setting the trend in the future [48]. In the medium or long term, the primary sectors in developed countries have serious profitability problems due to the drop in prices in the international agricultural markets and the increase in the costs of inputs, such as agrochemicals or the seeds themselves. Rural depopulation is the effect that warns about the rupture of the institutional framework of the original culture linked to the community vision of natural resources [49,50].

The extension of the productivity vision of agriculture in Latin America has been produced through public policies that, financed with international funds, have promoted investments in irrigation infrastructures that encourage intensification and specialization in crops. In other words, it seeks to maximize financial performance, even if it means putting sustainability, efficiency and equitability of water at risk, and displacing traditional diversified crops destined for food sovereignty [51]. In Manabí, crops such as coffee destined for world markets are displacing other crops such as beans, cassava, bananas or peanuts consumed by local communities.

The private management model is endangering family farming that is practiced in the Portoviejo River Basin. After the privatization of the system, intense conflicts were generated among users, since the interests, cultural roots and sustainability criteria of the participants were not taken into account. State intervention, associated with market interest, has negative effects on the sustainability of family production systems as a result of pressure exerted by power groups on systems that are successfully managed collectively as communal goods. In this stage, the informal social networks that stimulated development around the peasant culture weakened or were lost. With this, a water culture that is based on food security and sovereignty has been put at risk.

In the Portoviejo River Basin, the pressure to increase the cultivation areas and grasslands in the sensitive natural ecosystems of the area is highlighted. This causes the loss of the natural habitat of many endemic species of flora and fauna. Furthermore, the characterization of the region as a subtropical desert makes it especially vulnerable to climate change, which is occurring on a planetary scale, and which the government of Ecuador is incorporating in its policies [52]. The dry season is prolonged while the wet season produces heavy rains that increase the risks of floods and soil erosion in cultivated areas. In a market model, the adaptive capacity that smallholder cultures have been shown to have is lost. The reduction in profitability, per cultivated hectare caused by climate change, proposes as an alternative to the market, with the increase in agricultural areas and the intensification with agro-industrial techniques together with the consequent increase in water consumption, as well as polluting inputs. The preindustrial agriculture versus organic agriculture dilemma [53] must be posed by future research in the perspective of the analysis of institutions.

Agricultural activity has been strengthened with the transfer of the irrigation system to the Board of Users of the Rocafuerte Irrigation System [43]. In the context of the Poza Honda irrigation system, the market is a social construction that functions not only as a meeting place for sellers and buyers, but also for cultures in which the producers, besides exchanging goods and services, also exchange their production techniques and seeds. Producer decisions generally do not respond to traditional economic logic. Their rationality responds to a scale of values of respect for the land, water and nature. They display a diversity of forms of social organization that can be studied through the theory of institutions of collective action for the management of common goods.

This research determines, for the case study, how investments in the irrigation system to boost the productiveness of rural areas can also be the main cause of impoverishment of the social capital of these small farming communities and of the degradation of natural capital. The case study shows a correlation between the increase of a productive agriculture typology and the loss of social capital due to the displacement of better adapted cultural agro-ecological models mainly linked to family farming spread throughout the world [54]. However, this could be avoided by incorporating water

management models based on strategies that consider it as a common good and that take into account the institutional framework and procedures established in the community in accordance with their plans for good living. The results obtained from the case study are consistent with Ostrom contributions to water governance and that of scholars on the subject who have applied Ostrom framework to different cases in Latin America [55]. Likewise, some research paths that could be followed regarding this subject are proposed in order to give recognition to the small farming and indigenous culture based on the communal management of natural resources.

Faced with the market rationality that is so widespread and that is at the origin of the unsustainability of modern development, it is necessary to delve into other rationalities that incorporate the complexity of the institutional frameworks in which decisions are made. The importance of the immaterial elements of small farming cultures leads to the recognition of perception and subjective processes, which implies adopting a multidisciplinary approach. Water management as a communal good must be placed at the forefront of agricultural and sustainable development policy, taking into account the values of the ecosystem services associated with TEK, for which estimation methodologies are available and have not yet been incorporated. This research ultimately responds to its objective of giving evidence of the difficulty of stopping the processes that affect the organization of peasant cultures, as long as the recognition of the complex nature of common goods such as water is not deepened. Private water management creates irreversible socio-ecological risks and is the main cause of unsustainability.

**Author Contributions:** J.R. and B.V.C. developed Ostrom's methodology and applied it to the case study. Likewise, the authors collected, systematized and analyzed the information, and produced the first preliminary results for discussion. Both authors contributed to the Discussion section and to the writing and general edition of this research paper. All authors have read and agreed to the published version of the manuscript.

**Funding:** This research received no external funding.

**Acknowledgments:** We would like to thank the farming families, who generously provided the information needed through the interviews conducted. Likewise, we would like to thank the Universidad Laica Eloy Alfaro de Manabí (ULEAM) for providing the corresponding permits for the researcher to go to the research site.

**Conflicts of Interest:** The authors declare no conflict of interest.

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
