# Peer review of "Public Management, Private Management and Collective Action in the Portoviejo River Basin: Visions and Conflicts"

_sustainability, doi:10.3390/su12135467_

Round 1

Reviewer 1 Report

The paper is interesting and, in general, well structured in presenting the research. 

Below some suggestions to improve the paper (mostly addressed to improve the presentation clarity of method and results)

1) introduction - chapter 1 
the general introduction is well structured (I found very interesting the way in which authors combined the research theoretical framework/literature review and the main aspects related to the research case connected to the concepts presented). 
> a suggestion could be: adding a short paragraph (concluding the Introduction after row 156) of summary/recap of main research questions (that are mentioned in the text but it could be a useful support for the reading process include a short recap of R. questions at the end of the C1 and before to open the C2). 

2) materials and methods 
Step 1 and step 2 are clearly presented. I have one doubt (and a suggestion in order to clarify the presentation of the method) > the part described at rows 190 ("Secondary information from official sources was .....") is it a research "component/action" included in step 2 or is it a separated step (EG step 3)? the description of this research component is clear, interesting and an important component of the research design framework BUT in the paper is not clear to me if it is a specific step or it is a "crosscutting" component supporting the step 1 and step 2.

3. Results 
The paragraphs presenting the results (interesting) are clearly presented.

4 Discussion (and "general" notes) 
I have some questions (that could be also possible inputs for conclusions improvement): 

a) What are the impacts on ecological components/on natural resources (according to a long term perspective)
in my opinion, the presentation of the results could include also a specific focus on the impacts on ecological components deriving from the different water management models (ecological resilience and natural resources components).
if the research is not including this aspect (evaluation of impacts on ecological components, and it is perfectly fine!) it could be important to make it explicit in the research framework presentation (Reading the article, I was expecting a focus/evaluation on ecological components because the authors refer to Ostrom & socio-ecological systems)

b) I find some difficulties in understanding how the authors use the term "vision" (but it could be a personal limitation - I'm a planner - and as a designer, the "vision" is strictly connected/related whit the definition of strategies for sustainable "vision" or "scenarios" for territorial systems...).
we have two possible cases: 
a) the term "vision" is used for "water management models" (and it not require any "strategic ideas for the future")> it is fine.
b) the term vision is used ad "vision for future scenarios design". In this case, some proposal/perspectives to overcome the underlined phenomena could be important

About the conclusions/discussion... my feeling is that the paper could be richer in terms of "critical" outcomes is presented (EG it could be interesting to include a recap of main "positive/negative" impacts on socio-ecological systems) and in terms of "perspectives".
For example: focussing on the water issue it could be interesting also to add some considerations on Climate change and future scenarios (how the climate scenarios can change the existing conditions? and how local socio-ecological systems can be able to adapt/respond to local climate changes?).

Reviewer 2 Report

See file attached

Round 2

Reviewer 2 Report

The paper has been modified in depth. Authors have included the reviewer's suggestions